# Effect of Body Mass Index Percentile on Clinical and Radiographic Outcome and Risk of Complications after Posterior Instrumented Fusion for Adolescent Idiopathic Scoliosis: A Retrospective Cohort Study

**DOI:** 10.3390/jcm12010076

**Published:** 2022-12-22

**Authors:** Laura Scaramuzzo, Fabrizio Giudici, Giuseppe Barone, Pierluigi Pironti, Marco Viganò, Domenico Ravier, Leone Minoia, Marino Archetti, Antonino Zagra

**Affiliations:** 1Spine Surgery Division 1, IRCCS Istituto Ortopedico Galeazzi, 20157 Milan, Italy; 2Residency Program in Orthopedics and Traumatology, University of Milan, 20122 Milano, Italy; 3IRCCS Istituto Ortopedico Galeazzi, 20157 Milan, Italy

**Keywords:** adolescent idiopathic scoliosis, BMI%, posterior fixation, deformity

## Abstract

Background: The aim of this study is to evaluate the effect of body mass index percentile (BMI%) at postoperative and medium follow-up in AIS patients undergoing posterior instrumented fusion (PSF). Methods: We analyzed 87 clinical records of patients (19 male, 68 female) who underwent PSF. The patients were divided into four groups considering BMI%: underweight (UW), normal weight (NW), overweight (OW), and obesity (OB). Demographic, clinical (SRS-22), and radiographic data were collected. The primary outcome was to assess both the surgical and clinical outcomes, whilst the secondary outcome was to compare the radiological findings among the studied groups. Follow-ups were set preoperatively, at 6 months and 5 years. Results: Our results did not show significant differences of clinical outcomes among the studied groups, except for a longer surgical time and a higher hemoglobin decrease in UW and OB patients (*p* = 0.007). All BMI categories showed similar radiographic outcomes, with no statistical significance at final follow-up. OB patients showed a worse percentage of major curve correction compared to baseline and to UW and OW patients. Conclusions: The present study does not underline substantial differences in clinical and radiographic results among any of the studied groups. However, UW and OB patients showed a worse postoperative progress. Counseling should be provided for patients and families and the achievement of a normal BMI% should be recommended.

## 1. Introduction

Adolescent idiopathic scoliosis (AIS) is a common pathologic disease affecting 0.5% to 5% of young adolescents [1]. This deformity presents a multifactorial background, and its etiology is still under debate. Several factors are generally correlated to the spinal curve progression: curve magnitude and phenotype, peak growth velocity, gender, and skeletal maturity [2]. Recently published studies begin to also consider body composition as an important factor in AIS onset and development [3,4,5]. Indeed, body composition has been demonstrated to influence another important aspect of AIS: the outcome of both conservative and surgical treatments [6,7]. Despite several differences among the numerous studies in the literature, many of them have been conducted to only evaluate underweight (UW) adolescents, especially in the past decades, or obese (OB) adolescents, mainly in more recent years. This is due to the recent dramatic increase in the proportion of overweight (OW) adolescents [5]. Although anthropometric and body composition alterations are now recognized as being involved in the AIS course, the real relationships between different body mass index percentile (BMI%) types and both functional and clinical outcomes are not clearly understood [5].

Only a few studies have reviewed homogeneous populations and rarely are all ranges of BMI% compared at the same time in a single study. Upasani et al., in a retrospective analysis, report that OW adolescents with idiopathic scoliosis have greater thoracic kyphosis after surgical treatment [8]. Bjerke et al. report that both normal weight (NW) and OW patients reach equal surgical correction rates [9]. However, increased BMI% is correlated to major intraoperative blood loss with no increase of transfusion rates compared to NW patients, as stated by Hardesty et al. [6]. On the other hand, several studies point to the adverse effect of being underweight on surgical and functional outcomes [2]. Farahani et al. state that UW patients undergoing posterior instrumented fusion (PSF) show higher postoperative complication rates [10]. A BMI% lower than the 25th percentile for age is reported as a principal risk factor for superior mesenteric artery syndrome and postoperative acute pancreatitis [2,11]. Furthermore, the alteration of body self-perception in low BMI patients could negatively influence the surgical outcomes of AIS, despite the correction obtained [12].

The aim of the present study was to evaluate the effect of BMI% for all categories on postoperative and at medium follow-up in patients suffering from AIS undergoing PSF. The two BMI% distribution extremes, <5% and >95%, were compared to NW and OW. The primary outcome was to assess both the surgical and clinical outcomes, whilst the secondary outcome was to compare the radiological findings among the studied groups and the incidence of complications.

## 2. Materials and Methods

This retrospective, single center study was conducted in a single specialized center. The data were collected from patients included in the Institutional Registry (Spinereg Register). Institutional Review Board approval was obtained prior to the initiation of this study and was registered with number DSAN 854-B/3. The demographic, surgical, clinical, and radiographic data of 150 patients (68 males and 82 females) affected by AIS and surgically treated from January 2011 to December 2016 were collected. In this study, we included patients affected by AIS undergoing primary posterior instrumented fusion, aged ≤18 years old, with at least 5 years follow-up. The exclusion criteria were patients affected by either neurological or congenital scoliosis, previous spinal surgery, psychiatric disorders, and patients with severe chronic diseases (ASA > 2). All of the patients were subdivided into four groups considering BMI%: UW < 5%, NW 5% ≤ 84%, OW 85% ≤ 94%, and OB ≥ 95%. The primary and secondary outcomes of this study were respectively to assess both the surgical and clinical outcomes and to perform a radiological evaluation, comparing UW and OB patients with NW and OW patients. Surgical outcomes included level fused, estimated blood loss (BL) (in milliliters), surgical time (in minutes), and complications. Clinical and functional evaluation was performed using the Scoliosis Research Society (SRS)—22 questionnaire [13], consisting of five domains: Appearance, Activity, Mental Health, Pain, and Satisfaction (postoperative evaluation only). Radiographic evaluation consisted of preoperative long-cassette X-ray along with bending test, and postoperative long-cassette X-ray. In the coronal plane, major and compensatory Cobb angle, Lenke curve type [14], and Risser grade were evaluated, and thoracic kyphosis and lumbar lordosis were assessed in the sagittal plane. The percentage of correction of the major and compensatory curve were also calculated at each follow-up. Both clinical and radiological follow-ups were set preoperatively, at 6 months and 5 years.

### 2.1. Surgical Technique

Two senior surgeons with similar training performed all surgeries. All patients underwent posterior surgery under general anesthesia with spinal cord monitoring of somatosensory and motor evoked potentials. The patients were placed in the prone position on a radiolucent table.

Step 1. Subperiosteal dissection: After a standard midline incision, the subperiosteal dissection of the posterior soft tissues was performed. Before hook or screw application, an inferior facetectomy was performed at each level.

Step 2. Anchor point insertion: In the all-screw technique, titanium pedicel screws were inserted with the freehand technique with the assistance of C-arm fluoroscopy.(Figure 1).

In patients treated with the hybrid technique, pedicle screws were inserted in the lumbar and inferior thoracic region generally up to T10. In the upper thoracic region, pedicle hooks were positioned with a cephalad direction. At the superior end in the convex side, a transverse process hook with a caudal direction was positioned to obtain a stable claw construct. In both techniques, a terminal box at the superior and inferior end of the fusion area was included (Figure 2).

Step 3. Decortication: The laminae were thoroughly decorticated, the spinous process and the other spine constrains were removed in order to facilitate the correction maneuvers, and the bone graft obtained from decortication was used for fusion.

Step 4. Correction maneuvers: A first step of correction was performed by a previously countered rod into the reduction tabs, using the setscrews to reach the screw head. This way, a translation of the spine to the rod was obtained. After the rod was engaged in all anchors, the rod rotation instruments were attached to the rod and the surgeon, together with the assistant, performed a global de-rotation of approximately 90° in the direction of the concave side (Figure 3).

To obtain additional correction, especially when an axial correction was needed, a segmental de-rotation could be performed by inserting the second contoured rod on the convex side and then anchoring the screw set on the screw head to perform bilateral de-rotation. At the end of the correction maneuvers, the insides of the rods were examined and connected using two transverse connectors.

### 2.2. Statistical Analysis

The analyses were performed using SPSS software v25 (IBM, Armonk, NY, USA). The continuous variables were descripted either as mean and standard deviation or median and interquartile range in the case of normal and non-normal distribution, respectively, as determined by a Shapiro–Wilk test. The categorical variables were described as relative and absolute frequencies. Between-groups comparisons were performed with one-way ANOVA for repeated measures, while changes between groups over time were evaluated using two-way repeated measures ANOVA models. In the case of non-normal distribution, the Kruskal–Wallis test was used to test the differences among groups. Appropriate post hoc tests were performed to test pairwise difference in each subgroup. A chi-square test (or chi-square test for trend, if appropriate) was used to assess the differences between groups for categorical variables. Statistical significance was considered when *p* < 0.05.

## 3. Results

### 3.1. Patients’ Demographics

The data of 150 patients treated in the index period were reviewed. Radiological or clinical final follow-up data were missing for 37 (24.6%), 15 (10.0%) did not give their consent for the study, and 11 (7.3%) were lost to final follow-up. Then, the data from 87 patients were analyzed. Overall, the mean age was 14.66 ± 2.27 years old, with a large majority of females (n = 68) with respect to males (n = 19). The mean BMI for the entire cohort was 22.1 ± 5 kg/m²; 8 patients were UW, 55 NW, 14 OW, and 10 OB. The complete demographic data are summarized in Table 1. The age and gender distribution were similar among the different BMI groups.

### 3.2. Surgical and Clinical Outcomes

A summary of the clinical and surgical data is provided in Table 2.

The Risser grade and number of fused vertebrae were similar among the different BMI groups. The stratification for Lenke grade does not show significant differences in the four groups with a preponderance in all groups of Lenke 1 type (Figure 4).

On the other hand, significant differences were observed in terms of surgical time considering the four groups (*p* = 0.007, Kruskal–Wallis test) (Figure 5), with higher surgical time for UW and OB subjects. Pairwise comparisons showed a significant difference between NW and OB classes (*p* < 0.05) and a tendency for differences (*p* < 0.1) in UW vs. NW and UW vs. OW groups.

The UW and OB subjects showed a higher intraoperative BL compared to NW and OW patients, but the large individual variability prevented the observation of significant differences. On the other hand, the hemoglobin decrease showed significant differences among the BMI groups, with a decreasing trend for higher BMI (*p* = 0.007). Pairwise comparisons identified significant differences comparing UW subjects with OB patients (*p* < 0.01); tendencies (*p* < 0.1) for higher hemoglobin decrease were also observed in UW compared to OW patients and in NW compared to OB subjects (Figure 6).

The overall complication rate was 1.1%. No complications were registered in the OW or OB groups, while one and two cases were observed in UW and NW groups, respectively. Specifically, the single case in the UW group was pancreatitis, while the NW group showed a case of metabolic acidosis and a case of loss of lower limb motor-evoked potential. Despite an apparent negative association between the incidence of complications and higher BMI, this trend was not significant (*p* = 0.158).

Considering the patient-reported outcome measures, every sub-evaluation of the SRS22 test showed a significant improvement at final follow-up compared to baseline in each group. Significant improvements were also observed postoperatively. The final follow-up was significantly different from the postoperative scores in the NW subjects in the pain and function subscales, as well as in OB patients in the pain subscale.

The self-image analysis reported the most relevant improvements by comparing preoperative and 6 months postoperative follow-up and 5 years last follow-up results. Nevertheless, no significant differences between the groups were found, with similar improvements in all BMI categories. At the same time, no differences were observed between the BMI groups in terms of satisfaction score. Table 3 reports the summary results for each time point and each subgroup.

### 3.3. Radiographic Evaluation

The radiographic analysis included measurement of the major Cobb angle, compensatory Cobb angle, thoracic kyphosis, and lumbar lordosis preoperatively and at 5 years last follow-up. There were no significant differences in the Cobb angle changes among the different BMI groups, while significant differences were observed postoperatively and at last follow-up compared to baseline (Table 4). Concerning thoracic kyphosis and lumbar lordosis, no significant differences were observed among groups or postoperatively compared to baseline (Table 4).

Considering percentage changes in the correction angles (Table 5), no differences were observed among the groups for major Cobb % correction between the baseline and post-operative values, even if larger corrections were observed for the UW and OW groups compared to the NW and OB groups. No differences were observed for compensative Cobb angle correction and lumbar lordosis, while a significantly larger reduction in the thoracic kyphosis angle was observed in OW subjects compared to NW. The median percentage changes between the angles observed postoperatively and at last follow-up demonstrated an overall optimal maintenance of correction in all subjects. Small significant changes were observed in the OW patients, especially compared to those in the NW group (Table 5).

## 4. Discussion

BMI has been demonstrated to be a key factor in the surgical as well as the conservative management of adult idiopathic scoliosis, and most recently, its influence on the AIS course has received increasing interest in the literature [6,15]. However, concerning AIS, the number of studies on a single surgical approach evaluating a homogeneous cohort is still limited, especially when it comes to UW patients [16]. Furthermore, there is still no consensus about the real impact of BMI on AIS surgical outcomes, especially concerning either ends of BMI spectrum [6,7].

What emerged from the present study is that the hemoglobin decrease was greater in UW patients, and the surgical time was higher for UW and OB patients. The major hemoglobin decrease in UW patients could be due not only to the anemia itself, but also to a smaller blood volume and a different composure for coagulation factors.

The increased surgical time observed in the UW and OB group could be due to different factors. The presence of smaller or deformed pedicles (respectively in UW and OB) increases the time for screw positioning. The greater blood loss could also have a role in the surgical time.

Worse SRS-22 results for pain were observed in UW and OW patients compared to NW and OB patients (Table 3). In our opinion, in the UW group, pain is likely to be associated with a poor lean mass, worsened in the postoperative period by the surgical treatment. Furthermore, UW patients showed lower satisfaction score values at 6-month follow-up compared to the other groups. This difference overlaps at the 5-year follow-up. Radiographic evaluation showed no significant differences at baseline of the major Cobb angle, compensatory Cobb angle, thoracic kyphosis, and lumbar lordosis between the different BMI groups. Nevertheless, the correction percentage of the major Cobb angle was higher in the UW and OW groups compared to the NW and OB groups. In particular, OB patients showed the worst correction; this is probably due to a major curve stiffness.

Obesity currently represents one of the most important growing social issues among adults as well as children, and it has proven to be associated with several musculoskeletal conditions including AIS [6]. To clarify whether obesity could negatively affect the outcomes of spinal deformity correction surgery, Hardesty et al. conducted a retrospective study showing increased intraoperative blood loss, longer surgical times, higher use of crystalloids, and decreased ability to perform intrathecal analgesia [6]. Basques et al. show how an increased BMI can likely to be associated with adverse events, either mild or severe [17]. Moreover, de la Garza Ramos et al. state that a high BMI could represent a negative prognostic factor after long-segment fusion procedures for AIS [15]. Specifically, they demonstrate that, while NW and OW patients could achieve similar short-term outcomes, OB patients show significantly higher wound complication, readmission, and reoperation rates and longer hospital stays [15].

On the other hand, a high prevalence of low BMI has been associated with AIS, and a greater progression risk in that specific category seems to be conceivable [16,18]. However, its correlation with spinal deformity correction surgery remains poorly studied [16]. Tarrant et al. conducted a retrospective cohort study aiming to analyze the association between low preoperative BMI and the outcome of spinal fusion in patients suffering from AIS [16]. Although a higher preoperative coagulation abnormality rate and asthma incidence were observed, a greater thoracic curves correction rate was shown to be associated with low BMI [16]. In contrast, Tarrant et al. conducted a systematic review reporting worse surgical outcomes and lower satisfaction rate at 3 years follow-up among UW patients due to the more visible back and shoulder shape [2].

As reported in the medical literature, OB patients show worse curve angles at diagnosis both on the sagittal and the coronal plane [19]. The present study does not confirm the abovementioned statement. Otherwise, we found a lower Cobb angle percentage of correction among the OB compared to UW and OW patients. A possible explanation could be that our OB patient cohort underwent surgical treatment at an older age and with a comparable skeletal maturity. The presence of worse curve correction in these patients could be due to the unavoidable diagnostic delay along with joint overload and hormonal disbalance, leading to stiffer curves [20]. Moreover, it has also been shown that an abnormal body composition is likely to be associated with significantly lower body weight, BMI, lean mass, and bone mineral density, which could contribute to AIS etiology and progression [2,21]. Given the aforementioned considerations, it is feasible to assume the abnormal body composition as a cause of worse functional outcomes in both UW and OB patients after surgery.

Some authors have stressed the importance of rigorous pre- and postoperative nutritional care in AIS [2,22]. Indeed, a malnutrition universal screening tool (MUST) has been developed for malnourished and OB adults to obtain care plans [23]. Thus, we believe it is mandatory to apply the same concept in children suffering from AIS to minimize poor surgical outcomes and dissatisfaction rates, through strict presurgical food counseling for both patients and parents with the aim of attaining a normal BMI. OB patients can considerably reduce the risks and improve clinical outcomes even if they only reduce their weight to fall into the OW category.

To the author’s knowledge, this is the first study of a surgical approach in a single center with equal standards of care for all patients. Additionally, it is one of the few to consider the full range of BMI%, considering the subdivision between OW and OB patients in the outcome. However, some limitations must be acknowledged—first, its retrospective design and the small sample size, especially for UW and OB patients. Moreover, the absence of data about the BMI% at final follow-up, to better understand the incidence of the changes in BMI% on the final outcome, should be addressed.

## 5. Conclusions

In conclusion, the collected data do not underline significant differences in major and compensatory Cobb angle correction among the studied groups. In contrast, a larger reduction in the thoracic kyphosis angle in OW subjects compared to NW patients was observed. UW and OB patients undergoing PSF showed:Higher surgical time.Higher intraoperative blood loss.Major hemoglobin decrease.Longer recovery time.

Thus, considering the importance of appropriate perioperative nutritional care, strict presurgical food counseling is highly recommended and should be routinely included in surgical planning.

## Figures and Tables

**Figure 1 jcm-12-00076-f001:**
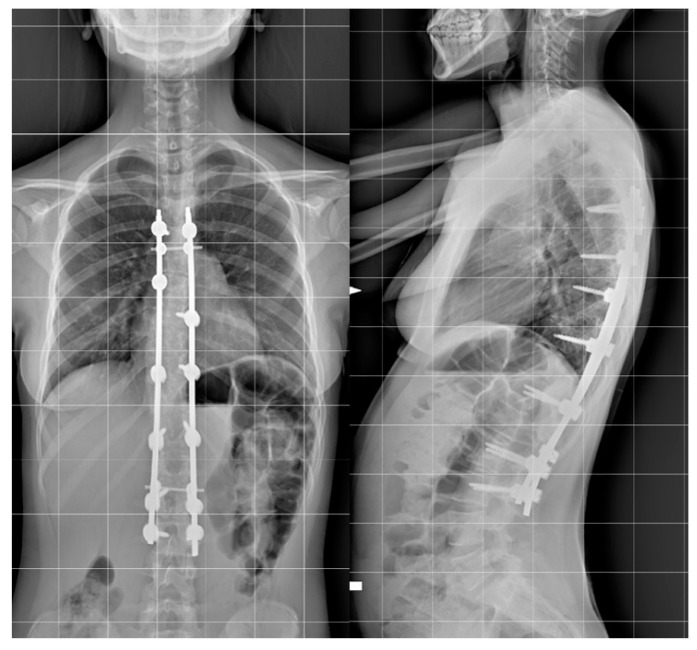
Example of all-pedicle screw construct for the treatment of AIS, AP, and LL.

**Figure 2 jcm-12-00076-f002:**
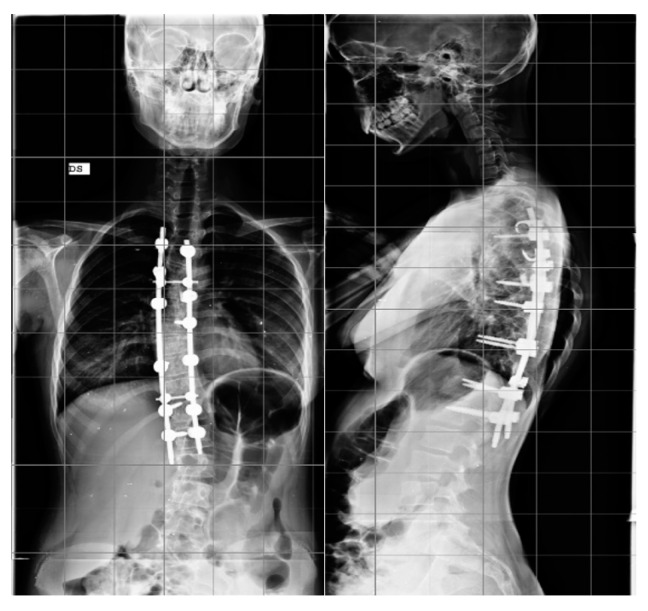
Example of hybrid construct for the treatment of AIS, AP, and LL.

**Figure 3 jcm-12-00076-f003:**
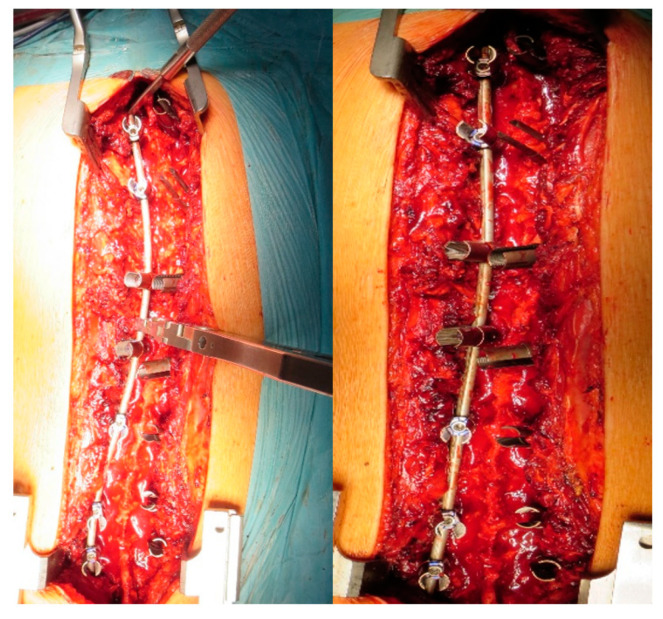
Intraoperative view of AIS correction maneuvers.

**Figure 4 jcm-12-00076-f004:**
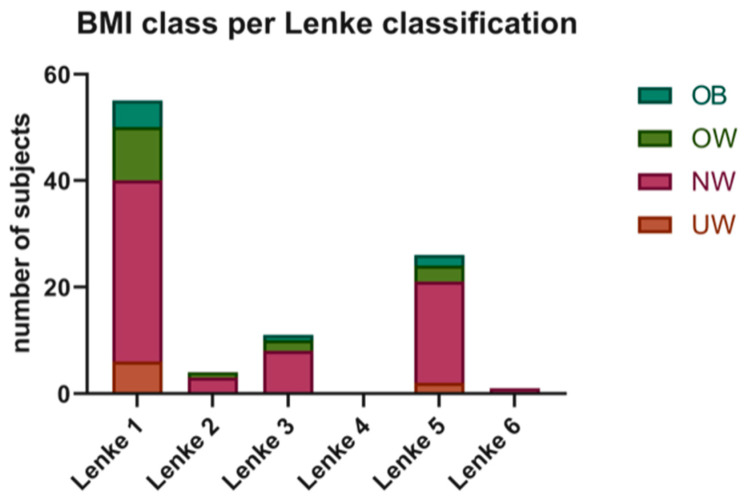
BMI classes (UW, BMI% ≤ 5; NW, 5 < BMI% ≤ 84; OW, 84 < BMI% < 95; OB, BMI% ≥ 95) per Lenke classification.

**Figure 5 jcm-12-00076-f005:**
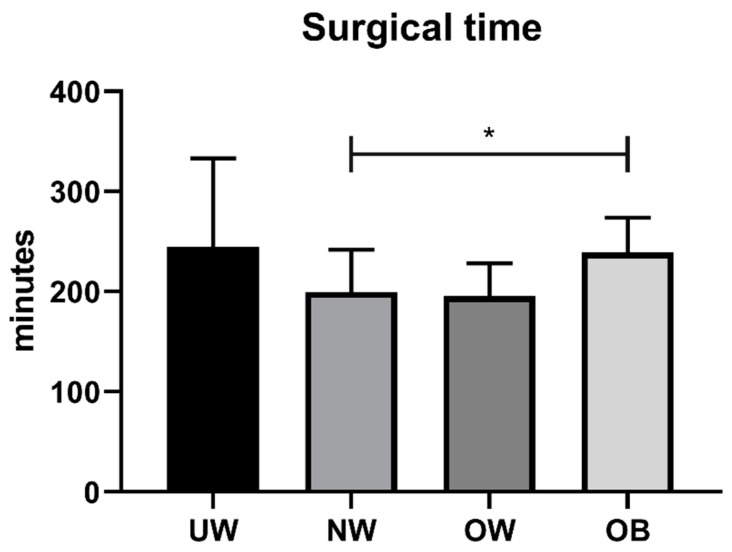
Surgical time in different BMI classes (UW, BMI% ≤ 5; NW, 5 < BMI% ≤ 84; OW, 84 < BMI% < 95; OB, BMI% ≥ 95). * *p* < 0.05.

**Figure 6 jcm-12-00076-f006:**
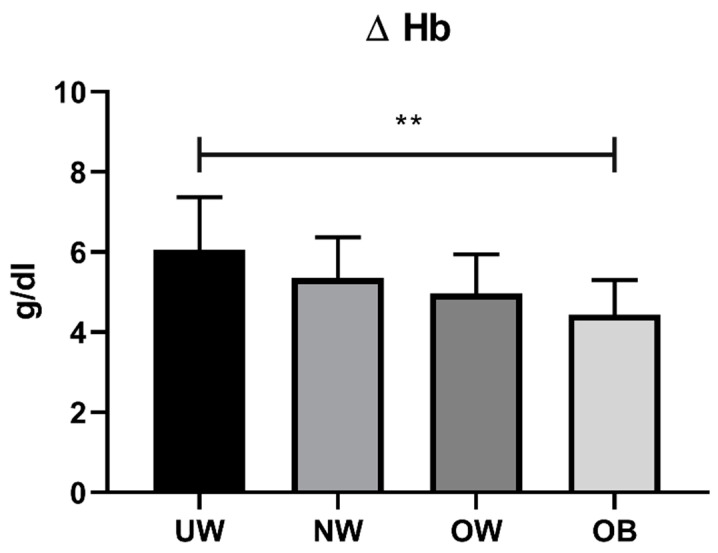
Hemoglobin loss in different BMI classes (UW, BMI% ≤ 5; NW, 5 < BMI% ≤ 84; OW, 84 < BMI% < 95; OB, BMI% ≥ 95). ** *p* < 0.01.

**Table 1 jcm-12-00076-t001:** Patients’ demographics.

	Underweight (BMI% ≤ 5)	Normal Weight (5 < BMI% ≤ 84)	Overweight (84 < BMI% < 95)	Obese (BMI% ≥ 95)	*p*-Value
N patients	8	55	14	10	
Mean Age	14.37 ± 2.07	14.84 ± 2.23	13.6 ± 1.65	15.5 ± 2.99	0.129
Gender	F: 7, M: 1	F: 42, M: 13	F: 11, M: 3	F: 8, M: 2	0.911

**Table 2 jcm-12-00076-t002:** Clinical and surgical data: IQR: interquartile range.

	Underweight (BMI% ≤ 5)	Normal Weight (5 < BMI% ≤ 84)	Overweight (84 < BMI% < 95)	Obese (BMI% ≥ 95)	*p*-Value
Patients (N)	8	55	14	10	
Mean surgical time (minutes)	244.62 ± 88.42	199.62 ± 42.23	195.64 ± 32.58	238.9 ± 34.85	0.008
Intraoperative blood loss (mL)	774.62 ± 365.79	627.35 ± 331.54	651.57 ± 244.77	765 ± 316.27	0.450
Delta Hb (g/dL)	6.05 ± 1.32	5.36 ± 1.01	4.96 ± 0.98	4.44 ± 0.86	0.007
Complications (N)	1	2	0	0	0.158
Fused vertebra (N)	11.5 (8–13)	10 (7–24)	11 (7–13)	10.0 (6–13)	0.225
Risser (median, IQR)	4 (2.75–4.25)	4 (3–4)	4 (2–4)	4 (3–4)	0.868

**Table 3 jcm-12-00076-t003:** SRS22 scores in the different BMI groups: * *p* < 0.05, ** *p* < 0.01, *** *p* < 0.001 vs. pre-op; ## *p* < 0.01, ### *p* < 0.001 vs. post-op. Statistical test: two-way ANOVA for repeated measures with post hoc pairwise *t* test with Bonferroni’s correction.

SRS 22 Domain	Underweight (BMI% ≤ 5)			Normal Weight (6 < BMI% < 84)			Overweight (85 < BMI% < 94)			Obese (BMI% ≥ 95)		
	Pre-op	Post-op	Last FU	Pre-op	Post-op	Last FU	Pre-op	Post-op	Last FU	Pre-op	Post-op	Last FU
Pain	22.2 ± 1.6	22.4 ± 0.5	23.4 ± 0.7	22.1 ± 1.6	22.6 ± 1.0	23.6 ± 0.8 ***###	22.4 ± 1.7	22.4 ± 0.6	23.4 ± 0.6 ***	22.2 ± 1.4	23.4 ± 0.7	24.1 ± 0.9 **##
Mental Health	21.0 ± 1.8	23.2 ± 1.0 **	23.5 ± 0.9 **	21.8 ± 1.9	23.1 ± 1.1 ***	23.6 ± 0.9 ***	21.3 ± 1.5	23.1 ± 1.5 **	23.6 ± 0.9 ***	21.8 ± 2.1	23.6 ± 1.1 *	24.2 ± 0.8 **
Self-Image	9.7 ± 2.7	23.0 ± 1.8 ***	23.6 ± 0.9 ***	11.2 ± 2.3	23.1 ± 1.6 ***	23.6 ± 1.1 ***	10.3 ± 2.1	23.1 ± 1.7 ***	23.6 ± 1.1 ***	12.2 ± 2.6	23.1 ± 1.4 ***	23.9 ± 0.7 ***
Function	21.1 ± 2.3	22.2 ± 1.4	23.7 ± 0.9 *	21.7 ± 1.6	22.7 ± 0.9 ***	23.6 ± 0.8 ***###	20.6 ± 1.6	22.9 ± 0.9 ***	23.8 ± 0.9 ***	21.3 ± 2.6	22.7 ± 0.8	24.3 ± 0.8 **
Satisfaction		8.6 ± 1.1	9.1 ± 0.6		8.9 ± 0.8	9.1 ± 0.7		9.0 ± 1.0	9.1 ± 0.7		9.0 ± 1.0	9.4 ± 0.7
Total	3.7 ± 0.3	4.5 ± 0.2 ***	4.7 ± 0.1 ***	3.8 ± 0.3	4.6 ± 0.2 ***	4.7 ± 0.1 ***##	3.7 ± 0.2	4.6 ± 0.2 ***	4.7 ± 0.1 ***	3.9 ± 0.3	4.6 ± 0.2 ***	4.8 ± 0.1 ***

**Table 4 jcm-12-00076-t004:** Radiologic evaluations. * *p* < 0.05, ** *p* < 0.01, *** *p* < 0.001 vs. pre-op. Statistical test: two-way ANOVA for repeated measures with post hoc pairwise *t* test with Bonferroni’s correction.

	Underweight (BMI% ≤ 5)			Normal Weight (6 < BMI% < 84)			Overweight (85 < BMI% < 94)			Obese (BMI% ≥ 95)		
	Pre-op	Post-op	Last FU	Pre-op	Post-op	Last FU	Pre-op	Post-op	Last FU	Pre-op	Post-op	Last FU
Thoracic Cobb angle (°)	53.7 ± 15.4	21.1 ± 12.3 ***	19.6 ± 10.3 ***	53.2 ± 15.9	23.0 ± 11.9 ***	22.35 ± 12.1 ***	60.8 ± 16.0	23.9 ± 12.5 ***	23.9 ± 13.1 ***	51.7 ± 23.7	21.4 ± 11.7 **	20.9 ± 11.0 ***
Lumbar Cobb angle (°)	39.2 ± 12.2	13.7 ± 8.6 ***	13.1 ± 8.2 ***	35.0 ± 14.4	11.9 ± 9.1 ***	11.6 ± 8.7 ***	45.0 ± 17.1	14.2 ± 8.2 ***	12.7 ± 13.3 ***	43.4 ± 13.4	12.6 ± 8.7 ***	12.3 ± 7.7 ***
Thoracic Kyphosis (°)	25.2 ± 14.3	23.1 ± 12.7	25.4 ± 12.8	26.4 ± 12.9	27.4 ± 9.2	28.1 ± 8.9	34.2 ± 19.7	27.7 ± 9.9	28.6 ± 10.3	34.8 ± 17.5	31.3 ± 17.5	32.2 ± 11.0
Lumbar Lordosis (°)	48.0 ± 17.9	45.7 ± 10.2	47.1 ± 9.8	55.1 ± 11.6	50.4 ± 8.5 *	50.8 ± 8.2	53.0 ± 10.3	47.8 ± 13.2	49.5 ± 15.5	61.2 ± 16.8	48.6 ± 13.9	48.6 ± 14.4

**Table 5 jcm-12-00076-t005:** Percentage change in correction angles expressed as median and interquartile range. ^†^ *p* < 0.05 vs. NW group. Statistical analysis: Kruskal–Wallis test with Dunn’s post-test.

% Correction	Underweight (BMI% ≤ 5)		Normal Weight (5 < BMI% ≤ 84)		Overweight (84 < BMI% < 95)		Obese (BMI% ≥ 95)	
	Post-op vs. Baseline	Last FU vs. Post-op	Post-op vs. Baseline	Last FU vs. Post-op	Post-op vs. Baseline	Last FU vs. Post-op	Post-op vs. Baseline	Last FU vs. Post-op
Thoracic Cobb Angle (%)	−62.0 (−78.0, −51.2)	0.0 (−1.9, 0.0)	−58.3 (−68.7, −51.1)	0.0 (−4.5, 0.0)	−64.6 (−71.0, −53.6)	0.0 (−2.6, 2.7)	−55.5 (−71.5, −49.2)	0.0 (−3.7, 0.0)
Lumbar Cobb Angle (%)	−65.7 (−75.1, −52.3)	0.0 (0.0, 0.0)	−72.1 (−77.7, −55.6)	0.0 (0.0, 0.0)	−74.4 (−81.7, −50.4)	−3.1 (−20.8, 0.0) ^†^	−59.7 (−82.4, −57.9)	0.0 (−10.3, 0.0)
Thoracic Kyphosis (%)	−10.2 (−31.8, +10.0)	+5.5 (0.0, +14.4)	+6.2 (−12.6, +30.9)	0.0 (0.0, 0.0)	−22.5 (−36.5, +8.8) ^†^	0.0 (0.0, 0.0)	−14.4 (−25.5, −0.7)	0.0 (0.0, +3.7)
Lumbar Lordosis (%)	−5.3 (−11.7, +3.4)	0.0 (0.0, +2.9)	−9.4 (−16.7, +4.7)	0.0 (0.0, 0.0)	−2.2 (−27.9, +9.1)	0.0 (0.0, +1.3)	−15.8 (−23.2, −10.3)	0.0 (−2.4, 0.0)

## Data Availability

The data that support the findings of this study are available in MDPI specific repository.

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
