# Peer review of "Effect of Body Mass Index Percentile on Clinical and Radiographic Outcome and Risk of Complications after Posterior Instrumented Fusion for Adolescent Idiopathic Scoliosis: A Retrospective Cohort Study"

_jcm, 2022, doi:10.3390/jcm12010076_

Round 1
Reviewer 1 Report
It is interesting to establish the relationship between BMI and AIS postoperative recovery. Authors should carefully study the comments and make improvements to the article step by step. After major changes can an article be considered for publication.
1. Is it better to use "adolescent idiopathic scoliosis" instead of "idiopathic scoliosis" for the first keyword?
2. The introduction should be segmented, and please try to discuss more with literature wherever possible to strengthen your introduction.
3. Abbreviations should follow the full name and be defined when they first appear, and only need to be defined once. Please check the full text..
4. Please add ethics committee and the approval code to the Materials and Methods section.
5. Is there a basis for the way BMI% is grouped? (lines 70-71)
6. Surgical technique was described extensively, so a schematic diagram should be added to help the reader understand.
7. Table 1 is incomplete, and all tables should be three-wire tables.
8. Figure 1 is extremely irregular, such as coordinate units, heading positions, etc. And there is also a lack of explanation, such as the relationship between Lenke classification and BMI groups.
9. What is the meaning of complication (n %) in Table 2? The readers of this article may not be clinicians, and it is recommended to add explanations to some medical terms or statistics method.
10. There are a lot of data in five tables, and the comparison and changing trend of data cannot be seen intuitively in this way. It is recommended to illustrate with some figures.
11. Conclusion is short and should contain more details of what was achieved in the study, perhaps use bullet points instead to better summarise the findings of your study.
Author Response
R: Is it better to use "adolescent idiopathic scoliosis" instead of "idiopathic scoliosis" for the first keyword?
A: Thank you for the comment. The keyword was modified in the text
R: The introduction should be segmented, and please try to discuss more with literature wherever possible to strengthen your introduction.
A: Thank you for your suggestion. The introduction section was revised according to your comment
R: Abbreviations should follow the full name and be defined when they first appear, and only need to be defined once. Please check the full text.
A: Thank you for the comment. A careful check of the abbreviation was done.
R: Please add ethics committee and the approval code to the Materials and Methods section.
A: Thank you for the comment. The information was add in the main text.
R: . Is there a basis for the way BMI% is grouped? (lines 70-71)
A: Thank you for the comment. The BMI% is grouped as recommended by the World Health Organisation
R: Surgical technique was described extensively, so a schematic diagram should be added to help the reader understand.
A: Thank you for your suggestion. Given the technique complexity, a schematic diagram could lead to an accuracy loss so we have chosen to adjust the surgical technique description in four phases and add some explicative figures in order to improve the reader understand.
R: Table 1 is incomplete, and all tables should be three-wire tables.
A: Thank you for the comment. Table 1 was modified according to the suggestion
R: Figure 1 is extremely irregular, such as coordinate units, heading positions, etc. And there is also a lack of explanation, such as the relationship between Lenke classification and BMI groups.
A: Figure 1 (Figure 4 in the latest version) has been modified to make it more regular. A sentence about Lenke classification has been included in the text for clarity.
R: What is the meaning of complication (n %) in Table 2? The readers of this article may not be clinicians, and it is recommended to add explanations to some medical terms or statistics method.
A: We apologize for indicating % while reporting only the number of complications as absolute frequency. Table has been modified. In addition, further information has been added to the table footnote in order to provide elements for table interpretation.
R: There are a lot of data in five tables, and the comparison and changing trend of data cannot be seen intuitively in this way. It is recommended to illustrate with some figures.
A: Figures about surgical time and Hb loss have been added to the paper.
R: Conclusion is short and should contain more details of what was achieved in the study, perhaps use bullet points instead to better summarise the findings of your study.
A: Thank you for your suggestion. The conclusion section was revised according to your comment
Reviewer 2 Report
The manuscript, entitled “Effect of body mass index percentile on clinical and radiographic outcome and risk of complications after posterior instrumented fusion for adolescent idiopathic scoliosis: a retrospective cohort study”, documents the effect of body mass index percentile for age on postoperative and at medium follow-up in AIS patients who underwent posterior instrumented surgery. Although small sample size prevented the observation of significant differences in some analyses, this manuscript is written clearly and rigorously. The present version raised several minor points to be justified before publication. The reviewer’s suggestions include the followings.
-Introduction, first and second paragraphs; Are there any differences for the following two sentences, “Body Mass Index Percentile (BMI%)” and “body mass index percentile for age (BMI %)”?
-I think there is no mention about the stratification for Lenke grade (Figure 1) in the text.
-The sentence ‘Although a higher preoperative coagulation abnormality rate and asthma incidence were observed, a greater thoracic curves correction rate was shown to be associated with low BMI.’ would need the reference number 14.
-A possible explanation should be further added for why hemoglobin decrease was greater in UW patients, and surgical time was higher for UW.
Author Response
R: Introduction, first and second paragraphs; Are there any differences for the following two sentences, “Body Mass Index Percentile (BMI%)” and “body mass index percentile for age (BMI %)”?
A: Thank you for the comment, no there are no differences between the two, is a writing error which has been revised in the text.
R: I think there is no mention about the stratification for Lenke grade (Figure 1) in the text
A: Thank you for the comment. The stratification for Lenke grade was inserted both in the method section and in the Results section.
R: The sentence ‘Although a higher preoperative coagulation abnormality rate and asthma incidence were observed, a greater thoracic curves correction rate was shown to be associated with low BMI.’ would need the reference number 14.
A: The number of references was inserted in the text.
R: A possible explanation should be further added for why hemoglobin decrease was greater in UW patients, and surgical time was higher for UW.
A: Thank you for the comment an explanation was inserted in the text.
Round 2
Reviewer 1 Report
1. I recommend that Figures 1, 2, and 3 be merged. Also, the size of the figure in Figures 2 and 3 should be normalized, as in Figure 1.
2. There are too many sections in the introduction, even a single sentence in a paragraph, which seriously splits the original meaning. Maybe you stretched the word "segment". The introduction is not simply divided into paragraphs, but should be divided according to the hierarchical relationship.
3. I suggest the authors to toroughly check the text for formatting mistakes and inconsistencies. (lines 103, 113, 126, 129, 132, 144, 218, 308, 378)
